# Peer review of "On fluctuating momentum-exchange in idealised models of air-sea interaction"

_Nonlinear Processes in Geophysics, 2019_

## Referee Comment (RC1) · Anonymous Referee #1 · 10 Sep 2019

In the present paper the properties of several linear models of idealized air-sea momentum exchange are discussed. Those models differ in the coupling type between air and sea variables, as well as in the forcing type. Analytical solutions for the covariances are presented and the energy budgets are discussed in terms of fluctuation dissipation relation. The fluctuation theorem is applied to the probability of energy fluxes.

I think the paper contains novel results, which are interesting in the context of modeling air-sea interaction. Using idealized models is one important way for approaching such a complex problem. I recommend the present paper for publication after the following comments are addressed.

Major comments:
In order to stress the relevance of the present work, I recommend in the introduction to discuss which features of the air-sea momentum interaction are captured by the linear models from the paper. The bulk parameterization is widely used in observational studies, but are there references for the linear model equations L1, L2 and L3 considered here? Several terms are neglected in these models, what is the justification. I understand that the linear form is analytically tractable, but for example the Coriolis term is missing in the model, which is also a linear term.

It seems to me that the L3 model corresponds to the model analyzed in Wirth 2018 and the models L1 and L2 are approximations to this model. This should be stated in the introduction. The differences of the present setup to the previous study of Wirth 2018 should be stressed.

Some models do not reach steady state. Can you give the growth rates in dimensional units for real air-sea configuration and estimate the relevance. In particular, some rates scale as $1/M^2$, where M is the total mass, suggesting very small growth. On this time scale probably other effects such as nonlinearity will become important.

Minor Comments

Abstract: replace:

The short term behaviour is similar, which ... with The short term behaviour is similar, with ...

p. 2 l.25: replace liaison with lesson

p.3. It is not clear to me the relevance of the 2D energy cascade dynamics discussed around line 20 to the present paper. Can you specify the connection with the work of Wirth 2018 and the lack of time scale separation between the forcing and atmosphere dynamics.

p.3. The analytic solution of linear model gives ... This sentence is confusing, rewrite.

p.3 It is shown in Wirth 2018, by solving the FPE ... Are you speculating that the linear models, discussed here, capture features of the nonlinear models? What is the exact connection between the linear/non-linear models in Wirth 2018 and the models discussed here.

p.5 eq. 1,2 Below it is stated that eq. 1, 2 represent a classical approach to implement air-sea interactions. Is this approach including only the linear eq.1,2 or other terms are included as well, can you give references.

p.5 below eq. 3,4 This model neglects the action of ocean currents, it is used ...

Can you provide references, are only eq.3,4 used in those studies?

p.7 paragraph above eq. 10: small omega is introduced to denote ensemble realizations. I found it confusing since it also denotes the frequency of the periodic forcing on p.6. line 6.

p.8 line 13: replace $\mu$ « SM by $\mu$ » SM

p.8 line 28: probably times t» SM**(-1) have to be considered.

p.9 line 15: A dot is missing before "The total energy ..."

p.9 line 18: It is not clear why exactly this is a double FDR, can you please explicitly give the quantities exactly related.

p.13 eq.20: $\chi$ is called response operator as well.

p.15. below eq. 23: When the interval ...., the FT holds when:

"When" appears two times in the sentence. Can you state under which conditions exactly the FT can be applied.

p.15. line 15: the time-averaged energy: probably time-averaged work?

p.18, line 7: The present calculations can be used to guide applications of the FDT to systems with large, but not infinite, time separation.

This I found very interesting conclusion, can you be more precise here. Sometimes the governing equations of the fast processes, hidden in the forcing are not known (or those processes have to be parameterized), will extending the phase space still work?

p.19, eq A4 replace: $ADA^{(-1)}$ with: $ADA^{(-1)}u$

Is P always diagonalizable in applications?

p.20 below eq. A12 replace $\exp(Dt)$ with $\exp(D \Delta t)$

same below eq. A19 and A26

p.24 Appendix B8 Is the definition of $u\_t$ and $u\_s$ required: the explicit form of $u\_a$ and $u\_o$ is given below anyway.

p.27 Title Appendix B11: change to L2W

p.28 Title Appendix B12: change to L3W

––––––––––––––––––––

---

## Referee Comment (RC2) · Anonymous Referee #2 · 11 Sep 2019

[referee-annotated manuscript omitted]

---

## Author Comment (AC1) · 14 Oct 2019

Dear Editor,

I am grateful to the reviewers for their thorough review which has helped to considerably improve the manuscript. I added a sentence to the acknowledgments to thank them.

Sincerely,

Achim Wirth

The reviewers' comments are reproduced in blue and my answers are written in black and the changes added to the manuscript are given in red.

**Answer to Reviewer # 1:**

Anonymous Referee #1

In the present paper the properties of several linear models of idealized air-sea momen- tum exchange are discussed. Those models differ in the coupling type between air and sea variables, as well as in the forcing type. Analytical solutions for the covariances are presented and the energy budgets are discussed in terms of fluctuation dissipation relation. The fluctuation theorem is applied to the probability of energy fluxes. I think the paper contains novel results, which are interesting in the context of modeling air-sea interaction. Using idealized models is one important way for approaching such a complex problem. I recommend the present paper for publication after the following comments are addressed.

Major comments:

In order to stress the relevance of the present work, I recommend in the introduction to discuss which features of the air-sea momentum interaction are captured by the linear models from the paper. The bulk parameterization is widely used in observational stud- ies, but are there references for the linear model equations L1, L2 and L3 considered here? Several terms are neglected in

these models, what is the justification. I under- stand that the linear form is analytically tractable, but for example the Coriolis term is missing in the model, which is also a linear term.

To put more emphasis on (linear) Rayleigh friction I added the reference that discusses Rayleigh friction in detail and also gives references to numerical experiments:

(see Stevens et al. (2002) for a detailed discussion and justification on using the linear Rayleigh friction).

The reviewer is right, the Coriolis term is omitted. An important point is that with a Coriolis parameter detailed balance is lost. I have started to think about the consequences but I am not sure about all the implications yet. The problem becomes similar to the one discussed by Speck and Seifert (2006). Conceptually the problem with a Coriolis parameter is different. Especially for the FDT further work along the line of Speck and Seifert (2006) is necessary (which possibly leads to substracting the inertial oscillations in the FDT analysis).

The major reason is, however, that, to my understanding, in this case a model of two interacting mixed layer models with vertical dependence has to be used. The dependence in the vertical direction is difficult to parameterize. This is a project that I am following for the moment in collaboration with Florian Lemarié (and which is funded by LEFE/CNRS this year) using his model of coupled mixed layers and comparing it to the simple models discussed in the present paper. For these models an analytic treatment is out-of-reach and numerical simulations for a statistically significant ensemble are expensive. Preliminary results suggest that the model behaves similarly to the model discussed here with respect to the FT. This work is mostly numerical and of a very different nature than the present work.

So I do perfectly agree with the reviewer and his comment is followed by ongoing work. I would also like to mention that recent analysis (to be submitted soon) suggests that results discussed in the present publication are also found observational data from satellites.

It seems to me that the L3 model corresponds to the model analysed in Wirth 2018 and the models L1 and L2 are approximations to this model. This should be stated in the introduction. The differences of the present setup to the previous study of Wirth 2018 should be stressed.

I added:

In the present work we consider models with an arbitrary forcing time scale and emphasise the differences of the one-way approximations to the two-way model introduced in (Wirth (2018)).

In the reminder of introduction it is stated that not only the FDR of the two way model is discussed but FDR, the FDT and the FT for all the models again emphasising the differences.

Some models do not reach steady state. Can you give the growth rates in dimensional units for real air-sea configuration and estimate the relevance. In particular, some rates scale as 1/M2, where M is the total mass, suggesting very small growth. On this time scale probably other effects such as nonlinearity will become important.

The reviewer is right, other processes take over to damp the growth, mostly non-linear horizontal turbulent dynamics. It was and is written in the discussion section: "In more involved models, divergence is avoided by other processes as non-linear interactions, increased horizontal dissipation or data assimilation, which drain energy in a different way."

I now added after this paragraph:

The magnitude of the constant growth rate is the typical growth rate of the ocean dynamics shortly after the turbulent forcing by the atmosphere has started and before dissipative processes develop to counterbalance it. It depends on the strength of the atmospheric forcing, its coherence in time and the thickness of the ocean (mixed-) layer. Processes that lead to a saturation of the growth are of various nature, space and time dependent and typically non-linear and intermittent.

Minor Comments

Abstract: replace:

The short term behaviour is similar, which ... with The short term behaviour is similar, with ...

Done.

p. 2 l.25: replace liaison with lesson

I would like to keep "liaison". One definition in (https://www.merriam-webster.com/dictionary/liaison) "a close bond or connection : interrelationship".

p.3. It is not clear to me the relevance of the 2D energy cascade dynamics discussed around line 20 to the present paper.

I now added:

This means that the energy dissipation is negligible in purely two-dimensional dynamics at high resolution and therefore no dissipation term parameterizing the horizontal friction within the layers is included in our models

Can you specify the connection with the work of Wirth 2018 and the lack of time scale separation between the forcing and atmosphere dynamics.

I now added:

In the present work we consider models with an arbitrary forcing time scale and emphasise the differences of the one-way approximations to the two-way model introduced in (Wirth (2018)).

p.3. The analytic solution of linear model gives ... This sentence is confusing, rewrite.

I now write:

The analytic solution of a linear model gives the dependence on all parameters, while in a non-linear model the parameter dependence has to be numerically evaluated for each parameter.

p.3 It is shown in Wirth 2018, by solving the FPE ... Are you speculating that the linear models, discussed here, capture features of the nonlinear models? What is the exact connection between the linear/non-linear models in Wirth 2018 and the models discussed here.

The reviewer is right in Wirth 2018, this is shown for the "two-way" model only. I now changed to:

It is shown in Wirth (2018), by solving the Fokker-Planck equation, that the second order moments of the two-way non-linear model can be reproduced by a two-way linear model using an eddy-friction approach with an eddy coefficient that is obtained analytically.

p.5 eq. 1,2 Below it is stated that eq. 1, 2 represent a classical approach to implement air-sea interactions. Is this approach including only the linear eq.1,2 or other terms are included as well, can you give references.

I now changed the sentence to:

In the L1 model the ocean velocities are not considered when the shear is calculated, this was commonly done in ocean simulations in the past.

It is hard to find a publication where it is explicitly stated that ocean velocities are not considered in the shear calculations. Everybody just did it. The awareness that this is a problem came only with Duhaut and Straub (2006), to the best of my knowledge.

p.5 below eq. 3,4 This model neglects the action of ocean currents, it is used ... Can you provide references, are only eq.3,4 used in those studies?

I added:

(or its nonlinear version)

and put a reference to Duhaut and Straub (2006) but any other ocean only simulation with a prescribed wind-field could be cited as the as the wind does not change.

p.7 paragraph above eq. 10: small omega is introduced to denote ensemble realiza- tions. I found it confusing since it also denotes the frequency of the periodic forcing on p.6. line 6.

In the periodic forcing I now use $\kappa$ everywhere.

p.8 line 13: replace $\mu \ll SM$ by $\mu \gg SM$

119  Done

120  p.8 line 28: probably times $t \gg SM**(-1)$ have to be considered.

121  Done

122  p.9 line 15: A dot is missing before The total energy ...

123  Done

124  p.9 line 18: It is not clear why exactly this is a double FDR, can you please explicitly give the
125  quantities exactly related.

126  I changed to:

127  This is a double fluctuation-dissipation relation: the dissipation and the fluctuation are related,
128  firstly, by the equal growth rate of their squares ($2t$ terms cancel) and secondly the constant terms
129  add up to $R/M^2$.

130  p.13 eq.20: $\chi$ is called response operator as well.

131  I added: , also called response operator

132  p.15. below eq. 23: When the interval ...., the FT holds when: "When" appears two times in the
133  sentence. Can you state under which conditions exactly the FT can be applied.

134  The sentence is now replaced by the precise mathematical definition:

135  The FT holds when:

$$S_{\overline{Z}^{\tau}}(z) = \sigma \tau z, \tag{1}$$

136  in the limit of $\tau \to \infty$.

137  p.15. line 15: the time-averaged energy: probably time-averaged work?

138  Done

139  p.18, line 7: The present calculations can be used to guide applications of the FDT to systems
140  with large, but not infinite, time separation.

This I found very interesting conclusion, can you be more precise here. Sometimes the governing equations of the fast processes, hidden in the forcing are not known (or those processes have to be parameterized), will extending the phase space still work?

I also think that this is an interesting point as I have seen published work using the FDT where, to my understanding based on the results of this paper, the FDT can NOT be applied directly due to the finite correlation time of the forcing, but the numerical results seem to indicate that the FDT applies anyway. Is it because the forcing time scale is small compared to the dominant dynamics or are there other reasons? The concept of extending the phase space is not new but rarely (never) mentioned in the climate community. In this sense it is different in a model to perturb the CO2 forcing in a climate model or to force the CO2 variable in a climate model. I would like leave as is as I do not have precise knowledge about other published applications but I do think that one should be more careful.

The non-equilibrium dynamics formalisme to perform the parameterisation suggested by the reviewer is called "Mori-Zwanzig-projection" a longterm goal of my research is to developpe a Mori-Zwanzig-projection of air-sea interaction.

p.19, eq A4 replace: $ADA(-1)$ with: $ADA(-1)u$ Is P always diagonalizable in applications?

Done. Yes, as long as $m \neq 0, 1$, which do not occur in applications and $\mu \neq S, Sm, SM$ there is no reason while the forcing time should be exactly one of these times.

p.20 below eq. A12 replace $\exp(Dt)$ with $\exp(D\Delta t)$ same below eq. A19 and A26

Done

p.24 Appendix B8 Is the definition of $u_t$ and $u_s$ required: the explicit form of $u_a$ and $u_o$ is given below anyway.

Now omitted.

Done

p.28 Title Appendix B12: change to L3W Interactive comment on Nonlin. Processes Geophys. Discuss., https://doi.org/10.5194/npg- 2019-40, 2019.

Thank you !

**Answer to Reviewer # 2:**

Anonymous Referee #2

The manuscript discusses the atmosphere-ocean interaction with some tools used in statistical physics., namely, the Fluctuation Dissipation Relation, the Fluctuation Dissi- pation Theorem and the Fluctuation theorem. This is a novelty in the field of geophysi- cal processes. The author present three different kinds of atmosphere-.sea interaction and he consider four types of forc- ing. That is, 12 different models. In my opinion the manuscript must be accepted but after a improvement of the english writing. In the present version , the manuscript is hard to read.

I have not lived in an English speaking country for the last 16 years and my knowledge of the English language is deteriorating. Furthermore I do not have funding to pay for a corrector of the English. I am truly grateful to the reviewer to have helped also in this respect.

1) In some case the paragraphs are so small and in my opinion it is possible to put as part of the previous paragraph.

I did merge paragraphs where I found it possible.

2) I enclosed in this revision a pdf file with some suggestion to improve the English writing

The paper contains a section in which a discussion is made of the 12 models. However there is no section devoted to drawn the main conclusions, for example to discuss the contribution of

the use of statistical mechanics tools to the state of art of the atmosphere-sea interaction and the limitations of a linear study of this interaction (which is non linear(

I added in the discussion section:

Statistical mechanics furthermore gives us to likeliness of extreme events.

And the paragraph:

The here presented concepts are not restricted to momentum transfer, but can also be employed to study heat exchange between the atmosphere and the ocean, or to other processes in the climate system with diverse characteristic time scales. Ongoing research is directed towards considering the concepts presented here in a hierarchy of models with increasing complexity and in observations. This research is of a different nature, numerical and observational and will be described elsewhere.

Please also note the supplement to this comment: https://www.nonlin-processes-geophys-discuss.net/npg-2019-40/npg-2019-40-RC2- supplement.pdf

I am grateful to this reviewer for the corrections, the corresponding changes are not represented in red in the manuscript but are almost all considered in the new version. I also added in the "Local models" section a reference to a paper that discusses in detail the use of linear Rayleigh friction:

(see Stevens et al. (2002) for a detailed discussion and justification on using the linear Rayleigh friction).

Thank you !

**References**

Duhaut, T. H., and D. N. Straub, 2006: Wind stress dependence on ocean surface velocity: Implications for mechanical energy input to ocean circulation. *Journal of physical oceanography*, **36 (2)**, 202–211.

Speck, T., and U. Seifert, 2006: Restoring a fluctuation-dissipation theorem in a nonequilibrium steady state. *EPL (Europhysics Letters)*, **74 (3)**, 391.

Stevens, B., J. Duan, J. C. McWilliams, M. Münnich, and J. D. Neelin, 2002: Entrainment, rayleigh friction, and boundary layer winds over the tropical pacific. *Journal of climate*, **15 (1)**, 30–44.

Wirth, A., 2018: A fluctuation–dissipation relation for the ocean subject to turbulent atmospheric forcing. *Journal of Physical Oceanography*, **48 (4)**, 831–843.

---

## Author Comment (AC2) · 14 Oct 2019

[revised manuscript text omitted]
 | $\dfrac{Sm}{\kappa^2+(Sm)^2}$ | 0 | $\dfrac{Sm}{\kappa^2+(Sm)^2}$ | $\dfrac{Sm}{\kappa^2+(Sm)^2}$ | 0 | 0 | 0 | 0 | 0 |
| L2P | $\dfrac{Sm}{\kappa^2+(Sm)^2}$ | 0 | $\dfrac{Sm}{\kappa^2+(Sm)^2}$ | $\dfrac{Sm}{\kappa^2+(Sm)^2}$ | 0 | 0 | 0 | 0 | 0 |
| L3P | $\dfrac{Sm}{\kappa^2+(SM)^2}$ | 0 | $\dfrac{Sm}{\kappa^2+(SM)^2}$ | $\dfrac{Sm}{\kappa^2+(SM)^2}$ | 0 | 0 | 0 | 0 | 0 |
| L1W | R | 0 | 1 | $\dfrac{m^2-1}{m^2}$ | $\dfrac{1}{m^2}$ | $\dfrac{1}{m^2}$ | 0 | $\dfrac{1}{m^2-1}$ | $\dfrac{1}{m^2}$ |
| L2W | R | 0 | 1 | 1 | 0 | 0 | 0 | 0 | 0 |
| L3W | R | $\dfrac{1}{M^2}$ | $\dfrac{M^2-1}{M^2}$ | $\dfrac{m}{M}$ | $\dfrac{m}{M^2}$ | $\dfrac{1}{M}$ | $\dfrac{1}{M^2-1}$ | $\dfrac{1}{m}$ | $\dfrac{1}{M+1}$ |
| L1C | $\dfrac{R}{\mu(\mu+Sm)}$ | 0 | 1 | $\dfrac{\mu(m^2-1)-Sm}{\mu m^2}$ | $\dfrac{\mu+Sm}{\mu m^2}$ | $\dfrac{\mu+Sm}{\mu m^2}$ | 0 | $\dfrac{\mu+Sm}{\mu(m^2-1)-Sm}$ | $\dfrac{\mu-Sm}{\mu m^2}$ |
| L2C | $\dfrac{R}{\mu(\mu+SM)}$ | 0 | 1 | 1 | 0 | 0 | 0 | 0 | 0 |
| L3C | $\dfrac{R(\mu+S)}{\mu^2(\mu+SM)}$ | $\dfrac{\mu+SM}{M^2(\mu+S)}$ | $\dfrac{\mu(M^2-1)+SMm}{M^2(\mu+S)}$ | $\dfrac{\mu m}{M(\mu+S)}$ | $\dfrac{m(\mu+SM)}{M^2(\mu+S)}$ | $\dfrac{\mu+SM}{M(\mu+S)}$ | $\dfrac{\mu+SM}{\mu(M^2-1)+SMm}$ | $\dfrac{\mu+SM}{\mu m}$ | $\dfrac{\mu+SM}{\mu(M+1)+SM}$ |

**Table 1.** Energy fluxes for $t \gg (SM)^{-1}, \mu^{-1}$. The last column is the efficiency in the system as it compares the energy growth in the system to the energy injection. Note that for $\mu \gg SM$, LCx converges to LWx if $R \to R\mu^2$.

| Exp. | $\langle u_a^2 - u_a u_o\rangle$ | $\langle u_a u_o - u_o^2\rangle$ | $\langle u_a^2 - u_o^2\rangle$ | $\langle (u_a - u_o)^2\rangle$ |
|---|---|---|---|---|
| L1P | 1 | $-\dfrac{S^2}{\kappa^2}$ | $1-\dfrac{S^2}{\kappa^2}$ | $1-\dfrac{S^2}{\kappa^2}$ |
| L2P | $\dfrac{\kappa^2}{S^2+\kappa^2}$ | 0 | $\dfrac{\kappa^2}{S^2+\kappa^2}$ | $\dfrac{\kappa^2}{S^2+\kappa^2}$ |
| L3P | 1 | 0 | 1 | 1 |
| L1W | $\dfrac{m-1}{Sm^2}$ | t-dep. | t-dep. | t-dep. |
| L2W | $\dfrac{m}{SM^2}$ | 0 | $\dfrac{m}{SM^2}$ | $\dfrac{m}{SM^2}$ |
| L3W | $\dfrac{M+1}{SM^2}$ | $\dfrac{1}{SM^2}$ | $\dfrac{M+2}{SM^2}$ | $\dfrac{M}{SM^2}$ |
| L1C | $\dfrac{(m-1)\mu-Sm}{Sm^2(\mu+SM)}$ | t-dep. | t-dep. | t-dep. |
| L2C | $\dfrac{\mu^2}{SM(S+\mu)(Sm+\mu)}$ | 0 | $\dfrac{\mu^2}{SM(S+\mu)(Sm+\mu)}$ | $\dfrac{\mu^2}{SM(S+\mu)(Sm+\mu)}$ |
| L3C | $\dfrac{SM+(M+1)\mu}{SM^2(SM+\mu)}$ | $\dfrac{1}{SM^2}$ | $\dfrac{2SM+(M+2)\mu}{SM^2(SM+\mu)}$ | $\dfrac{
[revised manuscript text omitted]